

# Assessing universality of DNA barcoding in geographically isolated selected desert medicinal species of Fabaceae and Poaceae

Aisha Tahir[1], Fatma Hussain[1], Nisar Ahmed[2], Abdolbaset Ghorbani[3] and Amer Jamil[1]

[1] Department of Biochemistry, Faculty of Science, University of Agriculture, Faisalabad, Pakistan
[2] Centre of Agricultural Biochemistry and Biotechnology, University of Agriculture, Faisalabad, Pakistan
[3] Department of Organismal Biology, Uppsala Universitet, Uppsala, Sweden

Corresponding authors
Aisha Tahir, aishatahir85@gmail.com
Amer Jamil,
profamerjamil@gmail.com,
amerjamil@yahoo.com

## ABSTRACT

In pursuit of developing fast and accurate species-level molecular identification methods, we tested six DNA barcodes, namely ITS2, $matK$, $rbc$La, ITS2+$mat$K, ITS2+$rbc$La, $mat$K+$rbc$La and ITS2+$mat$K+$rbc$La, for their capacity to identify frequently consumed but geographically isolated medicinal species of Fabaceae and Poaceae indigenous to the desert of Cholistan. Data were analysed by BLASTn sequence similarity, pairwise sequence divergence in TAXONDNA, and phylogenetic (neighbour-joining and maximum-likelihood trees) methods. Comparison of six barcode regions showed that ITS2 has the highest number of variable sites (209/360) for tested Fabaceae and (106/365) Poaceae species, the highest species-level identification (40%) in BLASTn procedure, distinct DNA barcoding gap, 100% correct species identification in BM and BCM functions of TAXONDNA, and clear cladding pattern with high nodal support in phylogenetic trees in both families. ITS2+$mat$K+$rbc$La followed ITS2 in its species-level identification capacity. The study was concluded with advocating the DNA barcoding as an effective tool for species identification and ITS2 as the best barcode region in identifying medicinal species of Fabaceae and Poaceae. Current research has practical implementation potential in the fields of pharmaco-vigilance, trade of medicinal plants and biodiversity conservation.

## INTRODUCTION

Many species of plants belonging to multiple families are catalogued as medicinal plants on the basis of the presence of specific chemical constituents and their effects on the biological systems (*Herrera et al., 2016*). Fabaceae and Poaceae are among the largest plant families having medically and therapeutically useful species all over the world (*Gao et al., 2010*; *Dashora & Gosavi, 2013*; *Wariss et al., 2016*). Ethnobotanical investigations revealed that *Crotalaria burhia* has antimicrobial, anti-inflammatory, wound healing,

and antioxidant properties (*Kataria et al., 2010*). *Acacia sp.* are used in tonics and for the treatment of dysentery, asthma, constipation, fever and gastric problems (*Ahmed et al., 2014*). *Indigofera sp.* have antioxidant property which is used in the treatment of infectious diseases, abdominal and spastic pain, and skin problems (*Rahman et al., 2017*). *Cenchrus ciliaris* has been reported anodyne, diuretic and emollient (*Hameed et al., 2011*; *Wariss et al., 2013*). *Cymbopogon jwarancusa* is reported as expectorant and used in treatment of flu, infections and epilepsy (*Ahmed et al., 2014*).

Local communities of far-flung areas of the country including Cholistan rely on herbal remedies without considering proper identification and documentation of valued medicinal species (*Mahmood et al., 2013*; *Ahmed et al., 2014*). Excessive harvesting of medicinal plants is not only a threat to biodiversity but also leads to intentional and unintentional adulteration in herbal products due to unavailability of actual species and economical constraints (*Sagar, 2014*) as well as misidentifications due to superficial resemblance among species (*Joharchi & Amiri, 2012*). Conventional methods for species identification rely on the morphology only that prove inefficient when specimens are morphologically more similar but belong to entirely different taxa. In order to avoid the misidentification and adulteration, a simple, rapid and reliable identification method is inevitable. Methods of species identification from integrated specimens to processed products demand the incorporation of modern techniques and tools specifically if morphological characters are insufficient or unavailable for correct species assignment to unknown specimens (*Gathier et al., 2013*; *Mutanen et al., 2015*; *Ghorbani, Saeedi & Boer, 2017*).

DNA barcoding is introduced in 2003 as a molecular based species identification tool by using a short, variable and standardized DNA region, the barcode (*Hebert et al., 2003a*; *Hebert, Ratnasingham & DeWaard, 2003b*; *Hebert & Gregory, 2005*). In order to meet the criteria of DNA barcode, a gene locus must possess enough species-level genetic variability, short sequence length, and conserved flanking regions (*Giudicelli, Mäder & Freitas, 2015*). Common DNA barcodes proposed for plants are plastidial *mat*K, *rbc*L, ITS, *rpo*B and *rpo*C1, the intergenic plastidial spacers (*trn*H-*psb*A, *atp*F-*atp*H and *psb*K-*psb*I) and the nuclear internal transcribed spacers that have been used singly or in combinations (*De Mattia et al., 2011*; *Saddhe & Kumar, 2017*). Owing to the strengths and limitations associated with each marker, *mat*K and *rbc*L are recommended as core barcode regions, which worked well with many of the plant groups. The need of supplementary barcodes arose due to comparatively lower discrimination success rate of *mat*K +*rbc*L than COI in plants and inefficient resolution in difficult plant taxa such as *Quercus* and *Salix*. Among supplementary markers, several constraints are reported in *trn*H-*psb*A such as premature sequence termination, presence of duplicated loci, and variable sequence lengths (100–1,000 bp) thus paving the way for nuclear DNA region, ITS2 which is a part of ITS, either as individual barcoding marker or supplementary region with core barcode for quick taxonomical classification in closely related species of wide range of taxa such as in Fabaceae, Lamiaceae, Asteraceae, Rutaceae, Rosaceae and many more (*CBOL Plant Working Group, 2009*; *Chen et al., 2010*; *Gao et al., 2010*; *Hollingsworth, Graham & Little, 2011*; *Pang et al., 2011*; *Balachandran, Mohanasundaram & Ramalingam, 2015*; *Wu et al., 2017*), hence tested in the current study as well.

The present study aims to re-evaluate the universality of commonly used DNA barcoding loci, ITS2, *mat*K, *rbc*La, ITS2+*mat*K, ITS2+*rbc*La, *mat*K +*rbc*La, and ITS2+*mat*K+*rbc*La by applying them on medicinal plants indigenous to harsh environment of Cholistan Desert for the first time. The objective was to barcode the species and to compare the discriminatory power of the standard barcode regions that will be an addition to the previous barcoding studies on Fabaceae and Poaceae, which were conducted on geographically different species and populations (*Gao et al., 2010*; *Wu et al., 2017*). Bioinformatics approach was practiced in the investigation for sequence analysis and barcode region evaluation.

## MATERIALS AND METHODS

### Plant material

A total of 30 specimens belonging to seven species of Fabaceae and three of Poaceae were included in this study. According to ethnobotanical survey (*Hameed et al., 2011*; *Ahmed et al., 2014*), all of the collected species are commonly used as medicinal plants in herbal formulations, but they are difficult to identify morphologically specifically in dried and processed form. Subfamilies of the species under consideration are not mentioned in this study. At least three individuals were sampled for each species from different locations of the Cholistan desert. All the specimens were identified taxonomically with the help of plant taxonomist Dr Mansoor Hameed at Department of Botany, University of Agriculture, Faisalabad using published flora and monographs (http://www.tropicos.org/Project/Pakistan). Voucher specimens are deposited at the Herbarium of Department of Botany, University of Agriculture, Faisalabad. The samples were collected from wild and locations that did not include any park or protected area of land, nor did the collection involve any endangered species.

### DNA extraction, amplification and sequencing

Total genomic DNA was extracted from specimens by grinding silica-gel dried-leaf tissue in liquid nitrogen, and then using the CTAB procedure (*White et al., 1990*). Total genomic DNA was dissolved in TE buffer (10 mM Tris–HCl, pH 8.0, 1 mM EDTA) to a final concentration of 50 ng/$\mu$l.

Polymerase chain reaction (PCR) amplification of ITS2 and *rbc*La regions was performed in 50 $\mu$l reactions containing 25 $\mu$l of 10% trehalose, 0.25 $\mu$l of Platinum Taq-polymerase (5 U/ $\mu$l), 2.5 $\mu$l MgCl$_2$ (50 mM), 0.25 $\mu$l dNTPs (10 mM), 5.0 $\mu$l reaction buffer (10X), 0.5 $\mu$l of each primer (10 $\mu$M), 8.0 $\mu$l of ddH$_2$O and 8.0 $\mu$l of template DNA. PCR amplification of *mat*K was performed in 50 $\mu$l reactions containing 14 $\mu$l of 20% trehalose, 1.2 $\mu$l Taq-polymerase (5 U/$\mu$l), 1.2 $\mu$l dNTPs (10 M), 5.5 $\mu$l reaction buffer (10X), 1.5 $\mu$l MgCl$_2$, 2.8 $\mu$l of each primer (10 $\mu$M), 1 $\mu$l of template DNA and 20.0 $\mu$l of ddH$_2$O. PCR products were examined by electrophoresis using 0.8% agarose gels. The PCR products were purified using FavorPrep$^{TM}$ PCR Clean-Up Mini kit and then were sequenced using the amplification primers.

All the DNA regions were sequenced by using the BigDye$^®$ Terminator v3.1 Cycle Sequencing Kit (Applied Biosystems, Inc., Foster City, CA, USA) according to the protocol provided in a GeneAmp PCR System 9700 thermal cycler. Quarter volume reactions were

prepared with 0.5 μl sequencing premix and a 3.2 μM final concentration for the primers. The other components were 5× sequencing buffer and 3–20 ng PCR template. Standard cycling conditions were used (30 cycles of denaturation (30 s @ 96 °C); primer annealing (15 s @ 58 °C); extension (4 min @ 60 °C). Cycle sequencing products were precipitated in ethanol and sodium acetate to remove excess dye terminators. Then they were again suspended into 10 μl HiDi formamide (ABI) before sequencing on an automated ABI 3130 *xl* Genetic Analyzer (ABI).

## Data analysis
### Editing and alignment of sequences
The software program Geneious R9.1 (http://www.geneious.com) was used to visualize, assemble and edit the sequence trace files. Consensus sequences were aligned with the MUSCLE (*Edgar, 2004*) plugin in Geneious R9.1. Alignments were then further refined by eye examination for resolving any gaps, insertions or deletion. Sequences were exported from Geneious R9.1 as aligned FASTA files for further single-barcode (ITS2, *mat*K, *rbc*La) and combination-barcode (ITS2+*mat*K, ITS2+*rbc*La, *mat*K+*rbc*La, ITS2+*mat*K+*rbc*La) analyses. Only those species were included in combination-barcode analyses that have triplets of sequences of each marker of combination. The discriminatory power for all regions was assessed at genus and species level by employing four analytical methods i.e., BLAST, the pairwise genetic distance method (PWG distance), the sequence similarity method (TAXONDNA) and phylogenetic-based method (Neighbor-Joining and Maximum Likelihood phylogenetic trees).

### Analysis by BLAST procedure
All the newly acquired sequences were queried via BLASTn (http://blast.ncbi.nlm.nih. gov/Blast.cgi) against the online nucleotide database and further deposited in GenBank. BLAST was used to evaluate the species-level identification power of three markers and their combinations in the study. Aligned sequences were searched in National Centre for Biotechnology Information (NCBI) database through BLAST procedure (*Altschul et al., 1990*). Top matching hit having the highest (>98%) maximal percent identity score was the criteria for successful conspecific/congeneric identification.

### Pairwise genetic distance analysis
For the pairwise genetic-based method, average of inter-specific and intra-specific distances were calculated for both families separately in MEGA6 (Molecular Evolutionary Genetics Analysis Version 6.0) program (*Tamura et al., 2013*, http://www.megasoftware.net) and TAXONDNA software using the Kimura-2-parameter (K2P) distance model to explore the intra- and interspecies variations. The pairwise intra- and interspecific distances were calculated for each species of both plant families. For each single and multilocus barcode, the minimum interspecific distance was compared with its maximum intraspecific distance for the detection of barcoding gap (*Meier, Zhang & Ali, 2008*; *Van Velzen et al., 2012*).

### Sequence similarity analysis
In the sequence similarity method, the species identification potential of all barcode regions was assessed by calculating the percentage of correct identifications identified with the "Best

**Table 1** Sequence characteristics of ITS2, *mat*K and *rbc*La in selected medicinal species of Fabaceae and Poaceae.

|  | Fabaceae | | | Poaceae | | |
|---|---|---|---|---|---|---|
|  | ITS2 | *mat*K | *rbc*La | ITS2 | *mat*K | *rbc*La |
| Universality of primers | Yes | Yes | Yes | Yes | Yes | Yes |
| Percentage PCR success (%) | 85 | 71 | 100 | 100 | 100 | 100 |
| Percentage sequencing success (%) | 100 | 100 | 100 | 100 | 100 | 100 |
| No. of species (No. of individuals) | 7(21) | 7(21) | 7(21) | 3(9) | 3(9) | 3(9) |
| No. of no sequence/singleton species | 1 | 2 | 0 | 0 | 0 | 0 |
| Aligned sequence length (bp) | 360 | 844 | 553 | 365 | 772 | 553 |
| Parsimony-Informative sites (bp) | 200 | 98 | 43 | 106 | 27 | 16 |
| Variable sites (bp) | 209 | 99 | 44 | 106 | 27 | 17 |
| Average interspecific distance (%) | 0.35 | 0.07 | 0.03 | 0.26 | 0.02 | 0.02 |
| Average intraspecific distance (%) | 0.02 | 0.00 | 0.00 | 0.00 | 0.00 | 0.00 |

Match'' (BM) and ''Best Close Match'' (BCM) tests in Species Identifier 1.8 program of TAXONDNA software (*Meier et al., 2006*). Three aligned datasets of sequences of Fabaceae, Poaceae, and Fabaceae+Poaceae were prepared to compare the candidate markers' efficacy in closely and distantly related taxa. K2P distance model was used in this analysis.

### Phylogenetic analysis

In order to assess whether species are recovered as monophyletic groups, phylogenetic trees were reconstructed in MEGA6 after appropriate model selection in the same software for each single and combination barcode for all the studied species of both families. The barcode markers were compared on the basis of conspecific monophyletic clusters and the nodal bootstrap support in neighbor-joining (NJ) as well as in maximum-likelihood (ML) statistical methods (*Tang et al., 2015*; *Xu et al., 2015*; *Zhang et al., 2015*).

## RESULTS

### Amplification, sequence analysis, and genetic divergence

The three commonly used barcoding loci performed differently in terms of universality for amplification and sequencing in both families. Amplification success is 85%, 71% and 100% for ITS2, *mat*K and *rbc*La respectively for Fabaceae and 100% for all regions for specimens of Poaceae. Overall aligned length of the three regions ranged from 360 bp (ITS2) to 844 bp (*mat*K) for Fabaceae and from 365 bp (ITS2) to 772 bp (*mat*K) for Poaceae. In this study, 18 sequences of ITS2, 15 of *mat*K, and 21 of *rbc*La were generated from family Fabaceae and 27 sequences (triplicate of each species with each region) from Poaceae. In addition, ITS2 had the highest percentage of parsimony informative sites i.e., 56% (Fabaceae) and 29% (Poaceae), followed by *mat*K i.e., 12% (Fabaceae) and 3% (Poaceae) and *rbc*La i.e., 8% (Fabaceae) and 3% (Poaceae) (Table 1). Out of total seven medicinal species of Fabaceae, *Prosopis cineraria* was not amplified with ITS2 while *Crotalaria burhia* and *Prosopis cineraria* both were not amplified with *mat*K.

While comparing the markers in both families, *rbc*La was the best at amplification and sequencing followed by ITS2 and *mat*K while ITS2 had the highest percentage of

variable and parsimony-informative sites and *rbc*La had the lowest. The average intra- and interspecific divergence values in three barcoding markers in both families ranged from 0.00 to 0.02 and 0.02 to 0.35 respectively. *rbc*La showed the lowest average intraspecific (0.00) and interspecific (0.02) divergence. While ITS2 showed the highest intraspecific (0.02%) as well as interspecific (0.35%) divergences. Average sequence divergence values for *mat*K was slightly more than *rbc*La but much less than ITS2 i.e., 0.00 for intraspecific and 0.07 for interspecific (Table 1). Multilocus barcodes were prepared by concatenation of single barcodes hence their characteristics corresponded to their counterparts with an altered species identification effect.

In total, we generated 81 sequences (27 of ITS2, 24 of *mat*K, and 30 of *rbc*La) in this study. All of them are included in the analysis as single- and combination-barcodes. Fifty six refined sequences and metadata of all the specimens are submitted to BOLD systems under the project named "DNA barcoding of medicinal plants of Pakistan (DBMPP)" as well as in GenBank.

### DNA barcoding gap assessment

The relative distribution of the frequencies of K2P distances was calculated for the three single and four combined loci for the selected species of Fabaceae and Poaceae families included in the study using TAXONDNA software, thus barcoding gap was identified for all the barcoding markers. Pairwise intra- and interspecific genetic distances showed similar overlapped pattern for *rbc*La, ITS2+*mat*K and *mat*K +*rbc*La while distances were narrow in case of *mat*K and ITS2+*rbc*La. ITS2 among single, and ITS2+*mat*K+*rbc*La among multilocus markers have distinct gap between pairwise intra- and interspecies genetic distance at 1% and 0.5% divergence respectively. The discrimination power of a barcoding region was considered effective if the minimum interspecies distance was larger than its maximum intraspecies distance. Figure 1 is the illustration of the observed patterns in ITS2, *mat*K, *rbc*La, ITS2+*mat*K, ITS2+*rbc*La, *mat*K +*rbc*La and ITS2+*mat*K+*rbc*La.

### Species identification using BLAST

*rbc*La came up with the highest percentage of genus level identification while ITS2 leaded at species-level identification among all single and combination barcodes. In this analysis, *Lasiurus scindicus* of Poaceae was an ambiguous sample among the collection because it did not match with expected genus or species with all three markers while *Cymbopogon jwarancusa* of Poaceae did not match with expected genus/species with ITS2 but identified with other two markers. Overall, *rbc*La was better at identifying unknown specimens up to genus level followed by *mat*K and ITS2 in both Fabaceae and Poaceae (Table 2).

### Best match (BM) and best close match (BCM) analysis

The potential of all barcoding regions for species identification accuracy was estimated by measuring the proportions of correct identifications using BM and BCM functions. Both tools evaluate the proportion of correct identifications through different comparisons of input DNA sequences. In the SpeciesIdentifier program of the TAXONDNA software package, each sequence is compared with all other sequences present in the dataset and

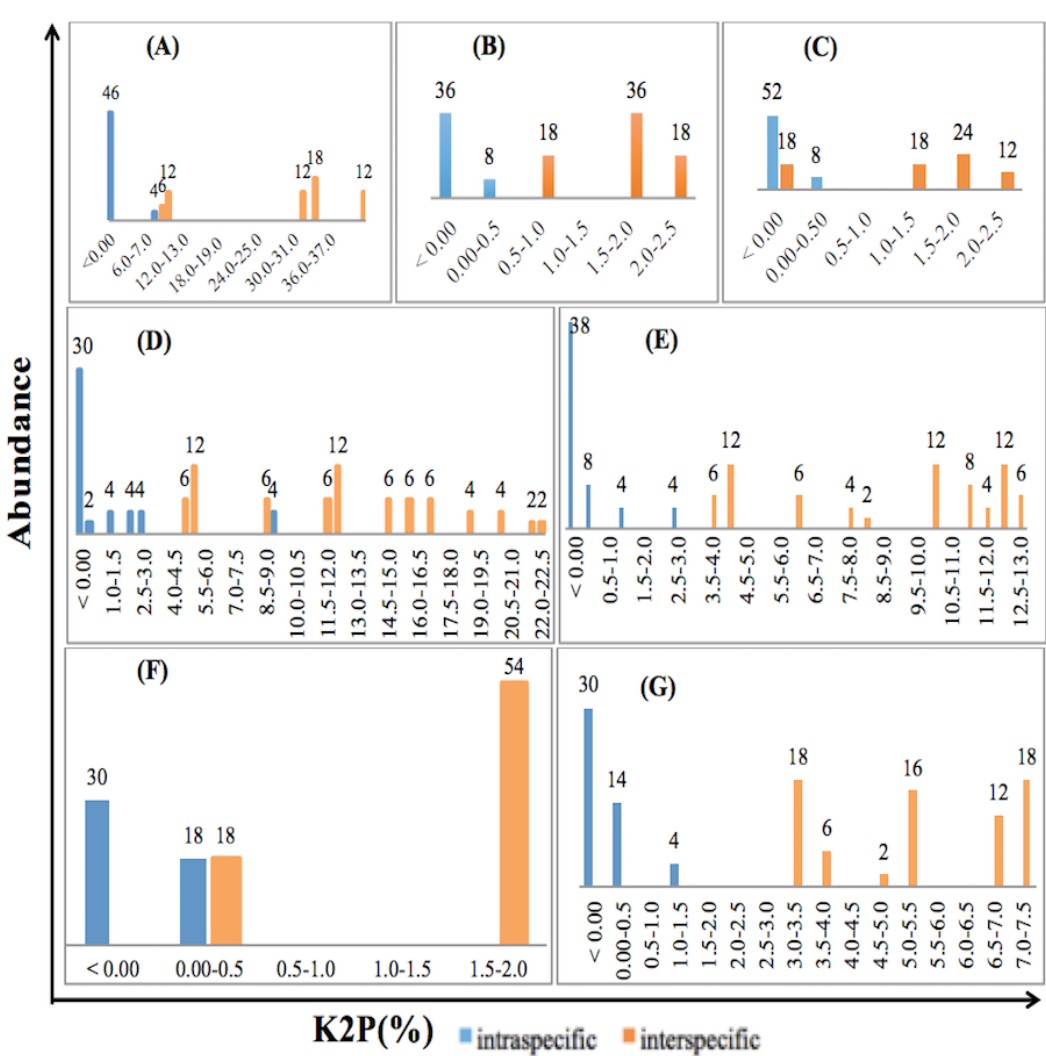

**Figure 1  Relative abundance of intra- and interspecific K2P pairwise distance for single and combination barcodes.** (A) ITS2. (B)*mat*K. (C) *rbc*La. (D) ITS2+*mat*K. (E) ITS2+*rbc*La. (F) *mat*K+*rbc*La. (G) ITS2+*mat*K+*rbc* La.

**Table 2  Genus and species-level identification success of candidate barcodes by BLASTn analysis.**

| Barcode region | Species-level identification rate | Genus-level identification rate |
|---|---|---|
| ITS2 | 40% (11/27) | 74% (20/27) |
| *mat*K | 37% (9/24) | 87% (21/24) |
| *rbc*La | 30% (9/30) | 90% (27/30) |
| ITS2+*mat*K | 37% (9/24) | 87% (21/24) |
| ITS2+*rbc*La | 22% (6/27) | 89% (24/27) |
| *mat*K+*rbc*La | 37% (9/24) | 87% (21/24) |
| ITS2+*mat*K+*rbc*La | 37% (9/24) | 87% (21/24) |

then compared sequences are grouped on the basis of their pairwise genetic distances that ultimately determines the conspecificity of two sequences.

The closest match of a sequence was established by BM function. Identification is categorized as correct if compared sequences were from same species and incorrect if the closest sequences were from different species. If a sequence matches with both the sequences i.e., of same species and of different species with equally significant similarity, then that sequence was considered ambiguous. The BCM function offered more stringent criteria by keeping a threshold of 0.1–0.5% pairwise distance in pairwise summary function. The queries above the threshold value were classified as "no match" and the others that are below the threshold value were analyzed according to the criteria established in "best match" analysis (*Meier et al., 2006*; *Giudicelli, Mäder & Freitas, 2015*; *Hartvig et al., 2015*; *Mishra et al., 2017*).

The results of sequence similarity test performed in TAXONDNA software for all single and combination barcodes are presented in Fig. 2. With both functions (BM and BCM), ITS2 was consistent in achieving the highest percentage of correct identification and the lowest number of unidentified sequences in all datasets. *rbc*La, showed the lowest discriminatory power for Fabaceae as six sample sequences were found ambiguous. An increase in identification power of *rbc*La is observed when it is combined with ITS2 in all datasets. "Incorrect" and "no match" were 0% in both functions so they are not shown in Fig. 2. This analysis indicates that the ITS2 met the rigorous standards for identifying the queries accurately among all single and combination barcodes.

### Tree based analysis of barcoding regions

Before proceeding to reconstruct the phylogeny using NJ and ML statistical methods, appropriate models having the lowest Bayesian Information Criterion (BIC) for the ITS2, *mat*K, *rbc*La, ITS2+*mat*K, ITS2+*rbc*La, *mat*K+*rbc*La and ITS2+*mat*K+*rbc*La were chosen (*Austerlitz et al., 2009*). Three types of observations were made in analysis of clustering pattern in all phylogenetic trees i.e., value of nodal support, clustering of species, family wise branching pattern.

Both, NJ and ML, statistical methods consistently recovered monophyletic clades at species level using all the single and combination barcodes except that of *rbc*La which could not discriminate between two species of genus *Acacia* of Fabaceae. Apart from tree topologies, bootstrap values were used as a criterion in this study, which was set at ≥ 99% as threshold. ITS2 under ML, ITS2+*rbc*La under NJ, and ITS2+*mat*K+*rbc*La under both phylogenetic methods worked equally well at species level for both families having higher percentage of nodes with ≥99% support as compared to other barcoding markers (Table 3).

## DISCUSSION

Floral biodiversity consists of a major category of medicinal plants that is important not only as a source of earning for local communities but also preserves traditional knowledge in the form of their medicinal uses (*Shinwari & Qaisar, 2011*). Our study approves the utility of DNA barcoding as species identification tool for the conservation of flora and

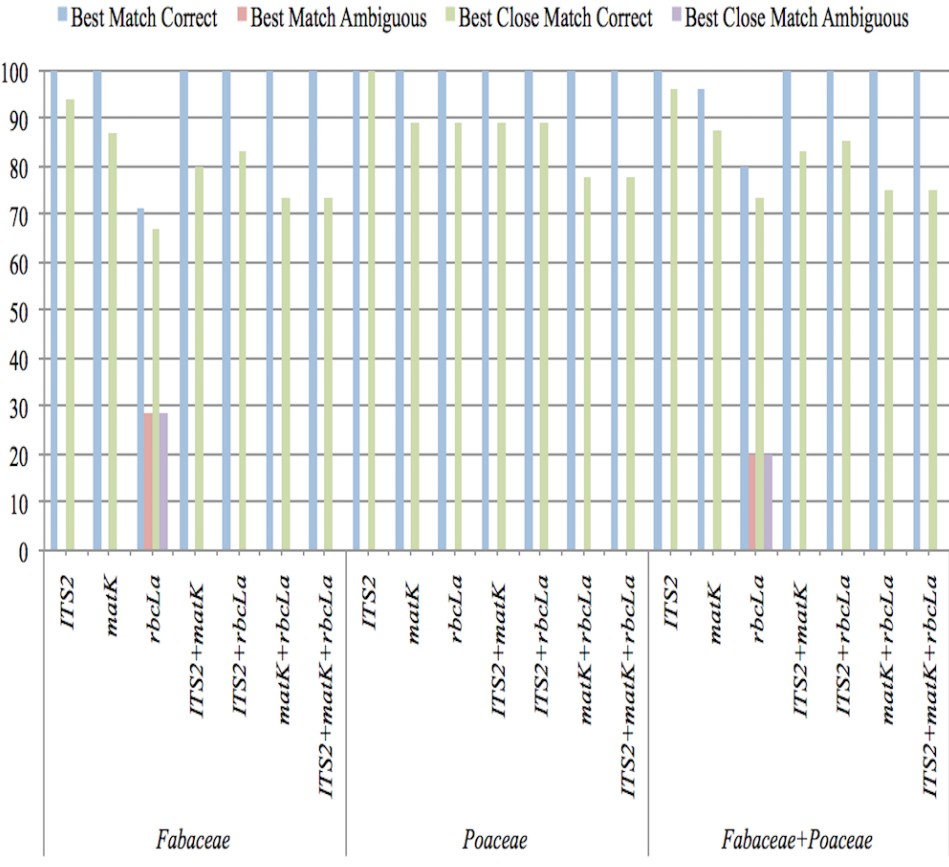

**Figure 2** Species-level discrimination ability of candidate barcodes by BM and BCM analyses.

**Table 3** Discriminatory power of single and combination barcodes based on phylogenetic trees.

| DNA barcodes | N[a] | Ability to discriminate (NJ)[b] (%) | | Ability to discriminate (ML)[b] (%) | |
|---|---|---|---|---|---|
| | | I | II | I | II |
| ITS2 | 27 | 87.50 | 25.00 | 100.00 | 88.88 |
| *mat*K | 24 | 100.00 | 0.00 | 100.00 | 25.00 |
| *rbc*La | 30 | 88.88 | 44.44 | 88.88 | 55.55 |
| ITS2+*mat*K | 24 | 100.00 | 75.00 | 100.00 | 55.55 |
| ITS2+*rbc*La | 27 | 100.00 | 88.88 | 100.00 | 50.00 |
| *mat*K+*rbc*La | 24 | 100.00 | 50.00 | 100.00 | 75.00 |
| ITS2+*mat*K+*rbc*La | 24 | 100.00 | 87.50 | 100.00 | 87.50 |

**Notes.**
[a]Number of nucleotide sequences.
[b]Column I: The percentage of conspecific monophyletic clusters. Column II: The percentage of conspecific monophyletic clusters with ≥99% bootstrap support value.
NJ, Neighbor Joining; ML, Maximum Likelihood.

safe use of medicinal plants of Fabaceae and Poaceae (*Gao et al., 2010*; *Saadullah et al., 2016*). Though, environmental (desert, marshes, lime rocks etc.) and biological factors (poorly dispersed, salt tolerant and relatively isolated species) influence the universality and standardization of DNA barcoding technique (*Yao et al., 2017*).

All of the barcoding regions included in this study are reasonably good regarding the universality in both families as reported earlier (*Yan et al., 2015*; *Li, Tong & Xing, 2016*). Since an ideal DNA barcode is expected to get amplified using standard PCR protocols in multiple species, we found that ITS2, *mat*K and *rbc*La fulfilled this criterion successfully with single pair of primers for each region. Comparatively, amplification success was slightly less for ITS2 and *mat*K than *rbc*La for Fabaceae that supports the opinion that barcodes are not consistent across the family Fabaceae but limited to a few genera (*Hollingsworth et al., 2009*). On the contrary, *Chen et al. (2010)* and *Han et al. (2013)* stated that ITS2 was relatively easy to be amplified using one pair of universal primers as well as ITS2 has also been reported for having ability to overcome the amplification and sequencing problems being shorter in length and conserved than ITS1 (*Yao et al., 2010*; *Gao et al., 2010*; *Pang et al., 2010*).

Sequence statistics determined that ITS2 had the most number of variable sites as well as relatively larger interspecific distance, the properties that strengthen a marker as ideal barcode region for its species discrimination ability (*Li, Tong & Xing, 2016*) that's why ITS2 is recommended as taxonomic signatures in systematic evolution (*Schultz et al., 2005*; *Coleman, 2007*). Core barcoding regions, *mat*K and *rbc*La also had variable, species specific informative sites but performed relatively poor than that of ITS2. In consistence with prior studies (*China Plant BOL Group, 2011*; *Zhang et al., 2015*; *Li, Tong & Xing, 2016*; *Saadullah et al., 2016*; *Mishra et al., 2017*), *mat*K and *rbc*La are recommended to be used as multi-locus barcodes (ITS2+*mat*K, ITS2+*rbc* La, ITS2+*mat*K+*rbc*La) as evident in Figs. 1 and 2 and Table 3.

Sequence analysis through BLAST and TAXONDNA determined that ITS2 identified the most number of specimens of both families at species level. Performance of *mat*K and *rbc*La was relatively weak at species resolution ability similar to the study of *Saadullah et al., (2016)* on the DNA barcoding of Poaceae. *rbc*La exhibited the highest genus level identification ability in both families. DNA barcoding gap also supported ITS2 region as a promising potential molecular marker to be used for species identification (*Li, Tong & Xing, 2016*).

Phylogenetic analysis provided a better species resolution than the nucleotide analysis (*Clement & Donoghue, 2012*; *Kim et al., 2016*) and has shown that despite of the fact that all of the barcoding regions except *rbc*La resolved specimens into distinct monophyletic clades at family, genus and species levels but considerably differed with respect to nodal support values. Phylogenetic trees of ITS2, ITS2+*rbc*La, and ITS2 +*mat*K+*rbc*La had similar percentage of nodes having 99% or more bootstrap support hence keeping the cost and time effectiveness into account, single barcode is preferred on multi-locus barcode specifically for small dataset (*Feng et al., 2015*; *Braukmann et al., 2017*; *Mishra et al., 2017*). This is in contrast to the study of *Hilu & Liang (1997)* and *Hollingsworth, Graham & Little (2011)* who have declared *mat*K as the best analogue of *CO1* animal barcode due to

rapidly evolving plastid DNA region. Phylogenetic analysis strengthens the application of DNA barcoding as the biodiversity conservation tool (*Hartvig et al., 2015*) and species authentication tool in quality control of herbal products (*Seethapathy et al., 2014*; *Vassou, Kusuma & Parani, 2015*).

## CONCLUSION

Based on the sequence statistics, inter- and intraspecific distances, BLAST, TAXONDNA and phylogenetic analyses, it is concluded that DNA barcoding is a rapid, convenient and universal species identification method that has been refined enough that it can discriminate the relatively isolated desert species as well as we suggest that ITS2 is the most suitable barcode markers for identification of medicinal species of Fabaceae and Poaceae.

## ACKNOWLEDGEMENTS

We are thankful to Dr. Mansoor Hameed for critical morphological authentication of plant materials and preserving them as vouchers in Herbarium, Department of Botany, University of Agriculture, Faisalabad, Pakistan.

### Funding
This work was supported by the Higher Education Commission Pakistan and the International Food Policy Research Institute (IFPRI). The funders had no role in study design, data collection and analysis, decision to publish, or preparation of the manuscript.

### Grant Disclosures
The following grant information was disclosed by the authors:
Higher Education Commission Pakistan.
International Food Policy Research Institute (IFPRI).

### Competing Interests
The authors declare there are no competing interests.

### Author Contributions

- Aisha Tahir conceived and designed the experiments, performed the experiments, prepared figures and/or tables, authored or reviewed drafts of the paper, approved the final draft.
- Fatma Hussain conceived and designed the experiments, contributed reagents/materials/analysis tools, authored or reviewed drafts of the paper, approved the final draft, collection of plant specimens.
- Nisar Ahmed, Amer Jamil and Abdolbaset Ghorbani conceived and designed the experiments, analyzed the data, authored or reviewed drafts of the paper, approved the final draft.

## Data Availability

Barcode of Life Data System:

http://www.boldsystems.org

DBMPP090-14, DBMPP091-14, DBMPP092-14, DBMPP089-14, DBMPP102-14, DBMPP103-14, DBMPP099-14, DBMPP100-14, DBMPP101-14, DBMPP109-14, DBMPP110-14, DBMPP111-14, DBMPP116-14, DBMPP117-14, DBMPP118-14, DBMPP180-14, DBMPP298-16, DBMPP299-16, DBMPP096-14, DBMPP097-14, DBMPP098-14, DBMPP106-14, DBMPP107-14, DBMPP108-14, DBMPP104-14, DBMPP105-14, DBMPP119-14, DBMPP113-14, DBMPP114-14, DBMPP115-14.

## Supplemental Information

Supplemental information for this article can be found online at http://dx.doi.org/10.7717/peerj.4499#supplemental-information.

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
