# Peer review of "Assessing universality of DNA barcoding in geographically isolated selected desert medicinal species of Fabaceae and Poaceae"

_PeerJ, doi:10.7717/peerj.4499_

## Round 0.1 · original submission · Major Revisions

· Academic Editor

Major Revisions

All three reviewers had some concerns about the manuscript. Please, supply an improved version addressing all comments and questions of the 3 reviewers. Please, describe the performed changes in an accompanying letter to the editor.

Reviewer 1 ·

Basic reporting

Line 121 “For each specie”, spelling error

Experimental design

1. Why should this work choose ITS2, matK and rbcLa? Have you ever tried the trnH-psbA region, which is regarded as a supplementary barcode to ITS2 for the identification of medicinal plants?
2. Have you ever tried the multi-locus barcodes in the scope of your study?Whether the identification efficacy would be better.
3. Table 1 and Table 2 should be displayed in the supplementary files or not described in the table way (e.g. direct explanation) in Materials and Methods part.

Validity of the findings

Research has been performed to use DNA barcoding for the identification of medicinal species of Fabaceae, and the result showed that ITS2 is an ideal region. What is the innovation and significance of your work?

Gao T, Yao H, Song J, Liu C, Zhu Y, Ma X, Pang X, Xu H, Chen S. 2010. Identification of
medicinal plants in the family Fabaceae using a potential DNA barcode ITS2. Journal of
Ethnopharmacology 130:116-121.

·

Basic reporting

So far, many researches are focused on finding universal plant DNA barcoding, and these DNA barcoding works very well only above the genus taxonomy. Therefore, they don’t focus on species identification in the case of very closely related taxa. This aspect of barcoding is extremely challenging and at the same time difficult to reconcile with barcode universality. This manuscript is focus on “DNA barcoding of selected desert medicinal species of Fabaceae and Poaceae” with a very good purpose.

Experimental design

However, very few samples were tested here to represent on different species. The samples here are representatives of different genera of Fabaceae and Poaceae.
There is not a sufficient number of individuals tested.

Validity of the findings

The manuscript title may be changed according to the plant material.
Please cite some recent references specifically working on species identification using DNA barcodes, like Wu et al., PloS one 12: e0182693.

A combination of ITS, matK, and rbcL should be tested in different genera of Fabaceae and Poaceae.
The indels need to be evaluated through different material.
The conclusion of the paper was “ITS2 and matK both are the suitable barcode markers for medicinal species of Fabaceae and Poaceae”. However, according to the results of BLASTn analysis, the species-level identification rates were low in all three regions. And as mentioned in Line 250-251, “Performance of matK and rbcLa was relatively weak at species resolution ability similar to the study of Saadullah et al. (2016) on the DNA barcoding of Poaceae.”, matK sequence didn’t work well in many species in Poaceae. The conclusion ‘matK was a suitable barcode marker for medicinal species of Fabaceae and Poaceae’ from the references was not true.
For the name of sequences, such as rbcL and matK, the last capital letter is not italic.
Line 283, 302 and 415: Reference with more than six author names, followed by “et al.”
Line 285, 301, 309, 409 and 416: The reference format of “PLoS ONE” was not consistent. Some are “PLoS ONE” and others are “PLoS One”.
Line 311: Italic.
Line 349: Italic.
Line 367: Italic.
Line 391: Italic.

Reviewer 3 ·

Basic reporting

Tahir et al. tested the efficiency of three DNA barcodes (ITS2, matK and rbcLa) for identification of frequently consumed medicinal species of Fabaceae and Poaceae indigenous to the desert of Cholistan using standard barcode gears. The paper was clearly written and well understood. The introduction part clearly showed the purpose of the paper, and cited important relevant literature. The structure of the paper complies with the journal standards, and raw fasta data was provided and solid. However, the tables and figures are not high quality according to criteria of standard publishing of this kind of work. Firstly, a three-line-table should be used; Secondly, all figures are low resolution and hardly readable; Last, bootstrap value of both ML and NJ method should be labelled, but not in %。

Experimental design

The research is in the scope of PeerJ, and the authors tried to evaluate the efficiency of three barcode markers in identifying medical plant of Fabaceae and Poaceae using a standard method. The questions and methods they undertook were well defined and described. However, I think the paper has severe drawbacks concerning selection of study species. Normally, we need to evaluate the utility of barcodes for discrimination of medical plants from its closest relatives and morphologically similar adulterants. Current research only sampled medicinal plants but not their closest relatives or adulterants.

Validity of the findings

The data were robust and statistically sound, and the conclusions were well stated. The work will add evidence for DNA barcoding as an effective tool for identification of medical plants.

Additional comments

The authors used three DNA barcodes (ITS2, matK and rbcLa) to identify ten medicinal species of Fabaceae and Poaceae in the desert of Cholistan using a standard barcode protocol. Although the paper was well written and structured, the methods were appropriate and results are solid, it is less meaningful concerning selection of study species, as no studied species’ closest relatives or adulterants are included in the analysis. Other concerns are listed below:
1) In comparing the inter and intra-species K2P distance, the authors should do it separately in Fabaceae and Poaceae, because if you choose species of two families to compare, the inter-species genetic distance should of course be larger.
2) Table 1 and Table 2 could be combined.
3) The NJ tree and ML tree could be combined.
4) The authors should state more clearly in the methods part why to choose these species to investigate. Are they all medicinal plants? Or, some of them are not medicinal plant and just be close relatives of studied medicinal plant. For example, the author only wrote in Introduction that Acacia nilotica is a medicinal plant, and how about A. jacquemontii and A. modesta?
5) A combination of all three markers should be assessed also by the standard barcode procedure, as normally more markers would do a better job in identifying species.
6) Some literature needs editing. For example, all genus names are not italic. I strongly recommend the authors check them throughout the paper.

---

## Round 0.2 · accepted · Accept

· Academic Editor

Accept

Your revised version of the manuscript has improved and can be accepted for publication.

·

Basic reporting

This manuscript is interested in selected desert medicinal species and to validate the potential barcodes for these medicinal species.

Experimental design

Combination of multi locus barcode has been analyzed.

Validity of the findings

The validated barcodes will be used for species discrimnation of these medicinal spcies.

Additional comments

This manuscript is significant improved after carefully revising by authors.